# Influence of Personality Traits and Organizational Justice on Job Satisfaction among Nurses

**DOI:** 10.3390/bs14030235

**Published:** 2024-03-14

**Authors:** Marin Mamić, Tihomir Jovanović, Slavka Galić, Ivana Jelinčić, Štefica Mikšić, Božica Lovrić, Ivanka Zirdum, Kristijan Matković, Goran Zukanović, Goranka Radmilović, Tihana Mendeš, Mirela Frančina, Ivan Vukoja

**Affiliations:** 1General County Hospital Požega, Osječka 107, 34 000 Požega, Croatia; mmamic@fdmz.hr (M.M.); bozica.lovric@pozeska-bolnica.hr (B.L.); ivanka.zirdum@pozeska-bolnica.hr (I.Z.); kristijan.matkovic@pozeska-bolnica.hr (K.M.); goran.zukanovic@pozeska-bolnica.hr (G.Z.); goranka.radmilovic@pozeska-bolnica.hr (G.R.); mirela.francinamf@gmail.com (M.F.); 2Faculty of Medicine, Josip Juraj Strossmayer University of Osijek, J. Huttlera 4, 31 000 Osijek, Croatia; tihomir.jovanovic@ozbpakrac-bhv.hr (T.J.); ivana.jelincic@kbco.hr (I.J.); tihana.mendes811@gmail.com (T.M.); 3Faculty of Dental Medicine and Health Osijek, Josip Juraj Strossmayer University of Osijek, Crkvena 21, 31 000 Osijek, Croatia; smiksic@fdmz.hr; 4General Hospital Pakrac and Hospital of Croatian Veterans, Bolnička 74, 34 550 Pakrac, Croatia; 5The Department of Social Sciences and Humanities, University of Slavonski Brod, 35 000 Slavonski Brod, Croatia; slavka.galic@pozeska-bolnica.hr; 6Department of Integrative Psychiatry, University Hospital Centre Osijek, J. Huttlera 4, 31 000 Osijek, Croatia; 7Faculty of Medicine, University of Rijeka, Ul. Braće Branchetta 20/1, 51 000 Rijeka, Croatia; 8School of Medicine, University of Zagreb, Šalata 2, 10 000 Zagreb, Croatia; 9Clinic for Otorhinolaryngology and Head and Neck Surgery, University Hospital Centre Osijek, J. Huttlera 4, 31 000 Osijek, Croatia

**Keywords:** job satisfaction, nurses, organizational justice, personality traits

## Abstract

The purpose of this research was to examine whether demographic variables, personality traits, and workplace variables (working in shifts, job tenure, and perceived organizational justice) contribute the most to the prediction of job satisfaction in nurses. The survey included 161 nurses. The instruments used in this research were as follows: the Demographic Data Questionnaire, the Perceived Organizational Justice Scale, the Job Satisfaction Scale, and the NEO five-factor inventory. The study findings indicated that age, health status, distributive justice, and procedural justice positively contribute to job satisfaction among nurses, while neuroticism contributes negatively. Older nurses, those in better health, those who are satisfied with the organization’s decision-making process, and those who feel adequately rewarded for their contributions tend to be more satisfied with their jobs. Conversely, nurses with a higher level of the neuroticism personality trait tend to be less satisfied with their job. The strongest predictors of job satisfaction among nurses were found to be health status, the personality trait of neuroticism, and distributive and procedural justice, with the age of nurses being slightly less powerful but still significant.

## 1. Introduction

Job satisfaction as a concept has been researched in many scientific disciplines. Interest in this aspect primarily originates from employers in the manufacturing sector, where the interest and intention is to create working conditions and an environment that will improve work productivity. When mentioning job satisfaction, the first thing that comes to mind is always the salary, especially from the point of view of employers, but sometimes also of workers. However, job satisfaction is much more complex and consists of several dimensions such as promotions, supervisor supervision, benefits, potential rewards, work activities, co-workers, work organization, and communication [1]. Several definitions or theories describe job satisfaction. Thus, it is defined as the mental attitude of an individual not only about the work environment, but also about family, health, and love, which indirectly reflects the work that the individual performs [2]. Other authors define it as a set of positive or negative feelings that employees have towards their work [3] or the attitude that workers have when their needs and interests coincide, when working conditions and rewards are satisfactory, and when they like to work with their colleagues [3].

In recent times, job satisfaction has been extensively researched, primarily due to its recognized importance in influencing employees’ health. It has been demonstrated that job satisfaction is one of the significant predictors of both physical and mental health, as well as subjective well-being [4,5]. Numerous studies have indicated that satisfied healthcare staff tend to be more efficient, productive, and loyal [6,7,8], making them a driving force for a healthy and productive organization [4]. However, the question remains: how can this be achieved? It is undoubtedly crucial, through research, to identify the factors influencing job satisfaction among nurses. In this study, we have identified several factors, both professional and personal, that could influence this construct. Through various research, it has been determined which factors directly contribute to nurses’ job satisfaction, or, when they are not met, to job dissatisfaction. These factors include working conditions, interpersonal relations, work organization, salary, job security, and working hours [6,9,10,11,12]. Nurses are expected to be good, kind, devoted to their work, dedicated, and helpful, which represents additional pressure for them. Additional pressure on their work is created by their already-chronic lack of time and resources. Daily encounters with patients whose basic needs sometimes cannot be met lead to moral dilemmas and are a source of frustration [13].

### 1.1. Literature Review

Several studies have revealed complex relationships between job satisfaction and a wide array of workplace characteristics, highlighting the multifaceted nature of employee satisfaction. One factor that has received substantial attention is organizational justice, with many researchers in recent decades demonstrating its crucial role in shaping perceptions of job satisfaction, including some conducted on a sample of nurses [14,15,16,17]. Organizational justice can be defined as the degree to which employees perceive that they are treated fairly in the organization across three dimensions: distributive, interactional, and procedural [14]. The research focused on identifying the dimension with the most significant influence on nurses’ job satisfaction and produced diverse outcomes. Regarding correlation, it demonstrated that job satisfaction is associated with all three dimensions of organizational justice [18,19]. Additionally, it was found among nurses that distributive justice had a more pronounced influence on job satisfaction than procedural justice and interactional justice in one study [20], whereas, in others, only distributive justice significantly contributed to nurses’ job satisfaction [21]. These findings emphasize the importance of organizational justice components regarding job satisfaction among nurses, but the inconsistency of the findings indicates a further need to study the relationship between these two constructs.

One of the factors that has proven significant in studies examining job satisfaction is personality traits. The relationship between personality traits and job satisfaction has been investigated across various professions, but research on this association among nurses is limited. The Big Five personality model includes dimensions such as extraversion, neuroticism, agreeableness, conscientiousness, and openness to experience [22]. When examining extraversion and job satisfaction, studies have consistently shown a positive association between these constructs [23,24,25]. In terms of neuroticism, research consistently indicates a negative relationship with job satisfaction [23,24,25,26,27,28], while conscientiousness has consistently demonstrated a positive association with job satisfaction [23,24,25,26,27]. Similarly, the agreeableness personality trait has shown a significant positive relationship with job satisfaction [23,24,25,27,29,30], and it has also proven to be a significant predictor of life satisfaction among nurses [21]. Regarding the personality trait of openness to experience, in most studies, it has not exhibited a significant correlation with job satisfaction [23,24], with only a few suggesting a weak positive association [27,31].

When discussing a potential explanation for the relationship between these constructs, a partial explanation can be found in the core self-evaluation model [21,32,33]. This theory determines one’s disposition towards job satisfaction, including self-esteem, general self-efficacy, locus of control, and neuroticism. An internal locus of control leads to higher job satisfaction, while lower levels of neuroticism also contribute to increased job satisfaction [34]. This theory could partially explain the influence of personality traits on job satisfaction.

As previously noted, there is a paucity of research on the relationship between these constructs within a sample of nurses. This presents an intriguing opportunity to explore whether a reciprocal relationship exists in a profession characterized by numerous interpersonal interactions, empathy, and assistance to others.

### 1.2. Aims

1. To examine the correlation between demographic (age, gender, and marital status), personal (personality traits), and job-related variables (job tenure, employment status, and perceived organizational justice) with job satisfaction among nurses and technicians.

2. To examine whether demographic, personal, and job-related variables contribute the most to the prediction of job satisfaction in nurses.

## 2. Materials and Methods

### 2.1. Population and Sample

A cross-sectional study was conducted. This research was conducted in the General County Hospital of Požega in the whole of 2023. Nurses of all levels of education participated in this research (high school, bachelor’s in nursing, graduate nurses, and master’s in nursing). This study included 165 nurses employed in the General County Hospital of Požega. From the total number, 161 nurses completed all the questionnaires and were included in the study, with 136 (84.5%) women and 25 (15.5%) men, with mean age values of M = 38.683 (SD = 12.531) and length of work experience M = 18.102 (SD = 12.959). The sample size was calculated using G*Power 3.1.9.7 software, which indicated that 143 subjects could provide a power of 0.8, with an effect size of 0.15 and 16 predictors at a significance level of 0.05.

### 2.2. Methods

The researcher distributed the questionnaires to the entire shift in a specific department collectively. Participants filled out the questionnaires in groups but were sufficiently separated, independently, to ensure privacy during completion. The interviewer was present nearby to provide assistance with any potential uncertainties. Once completed, the respondents placed the questionnaires in a separate folder, while the informed consent forms were stored separately, mixed with others, to ensure anonymity.

Participants were recruited based on the following inclusion criteria:
–Nurses who were employed at the institution where the research was conducted.

However, the exclusion criteria were as follows:
–Nurses who refused to participate;–Conditions that would hinder the completion of a comprehensive assessment, such as language barriers;–Lack of a signed informed consent form.

### 2.3. Materials

The following instruments were used in the research: the Demographic Data Questionnaire, the Perceived Organizational Justice Scale, the Index of Job Satisfaction, and the NEO five-factor inventory.

–Demographic Data Questionnaire

The questionnaire was developed for this study and includes information on age (participants provided their age), gender (male/female), marital status (married, divorced, single, widowed), employment status (permanent or fixed-term contract), job tenure (participants indicated their years of work experience), and health status (participants responded on a 5-point Likert scale, where “1” is very poor and “5” is very good).

–Perceived Organizational Justice Scale

The scale includes 20 items related to the experience of organizational justice, divided into three subscales (examining three dimensions of organizational justice): procedural, distributive, and interactional justice [14]. The result is the sum of items for each dimension of justice separately, and they are displayed separately. Cronbach α = 0.91 for the subscale of distributive justice, Cronbach α = 0.85 for the subscale of procedural justice, and Cronbach α = 0.88 for the subscale of interactional justice [14,35].

–Index of Job Satisfaction

The Index of Job Satisfaction is a five-item scale used as a measure of overall job satisfaction (Job Satisfaction Index) (36]. This scale assesses an individual’s general attitude towards their job (e.g., “I am quite satisfied with my current job” or “I enjoy my job”). The total score is the sum of scores on 5 items (of which items 3 and 5 are reverse-scored). The theoretical range of scores is 5–25, and its reliability expressed by Cronbach α = 0.88 [36,37].

–Neuroticism Extraversion Openness Five-Factor Inventory (NEO FFI)

The NEO-FFI inventory consists of 60 items designed to examine five major personality traits: neuroticism (Cronbach α = 0.84), extraversion (Cronbach α = 0.72), openness (Cronbach α = 0.58), agreeableness (Cronbach α = 0.66), and conscientiousness (Cronbach α = 0.80) [38,39,40].

### 2.4. Data Analysis

Continuous variables are summarized as the mean, range, and standard deviation. Categorical variables are summarized as numbers and percentages. Spearman’s correlation was utilized to examine the correlation, while the Point Biserial correlation was used for associations with dichotomous variables. Multiple hierarchical regression analyses (enter method) were performed to examine the contributions of independent variables to the explanation of dependent variables. All linear regression assumptions were met. The Kolmogorov–Smirnov test was used to test the normality of the distribution. A *p*-value < 0.05 was considered statistically significant.

Data were analyzed using JASP, version 0.17.2.1 (Department of Psychological Methods, University of Amsterdam, Amsterdam, The Netherlands).

## 3. Results

The results showed that the mean of life satisfaction among nurses was M = 18.885 (SD = 3.657) (Table 1).

Considering demographic variables, job satisfaction is weakly positively associated with health status (rho = 0.155; *p* = 0.049). In terms of personality traits, job satisfaction is weakly positively correlated with agreeableness (rho = 0.158; *p* = 0.045) and conscientiousness (rho = 0.156; *p* = 0.048) and weakly negatively correlated with neuroticism (rho = −0.219; *p* = 0.005) (Table 2).

Considering job-related variables, it has been found that job satisfaction is moderately positively correlated with distributive justice (rho = 0.407; *p* < 0.001) and procedural justice (rho = 0.333; *p* < 0.001) and weakly positively correlated with interactional justice (rho = 0.156; *p* = 0.049) (Table 3).

The results indicate that there are correlations between personality traits and organizational justice. Neuroticism demonstrates a low negative correlation with all three dimensions: distributive (rho = −0.189; *p* = 0.016), procedural (rho = −0.231; *p* = 0.003), and interactional (rho = −0.225; *p* = 0.004). Extraversion shows a weak positive correlation with distributive (rho = 0.226; *p* = 0.004) and procedural justice (rho = 0.251; *p* = 0.001), while agreeableness exhibits weak correlations with the distributive (rho = −0.238; *p* = 0.002), procedural (rho = −0.166; *p* = 0.035), and interactional (rho = −0.178; *p* = 0.024) dimensions (Table 4).

To identify the demographic, personal, and workplace-related variables that significantly predict job satisfaction, we conducted a hierarchical regression analysis. In the initial step, we included demographic variables. The demographic variables in this step explained 6.9% of the variance in job satisfaction (AR^2^ = 0.069). In the second step of the analysis, personality traits were added, revealing that the predictor variables in this step statistically contributed significantly to the explanation of job satisfaction and explain 14.9% of the variance (AR^2^ = 0.149). In the third step, variables related to the job were added. The predictor variables in this step significantly contributed to explaining job satisfaction, accounting for 27.7% of the variance (AR^2^ = 0.277). Significant predictors of job satisfaction among nurses were found to be age, health status, and neuroticism, along with distributive and procedural justice. Upon examining the coefficient of determination, it is evident that variables related to the job contribute the most to job satisfaction. The examination of the β coefficients indicates that age, health status, distributive justice, and procedural justice positively contribute, while neuroticism negatively contributes to job satisfaction among nurses (Table 5).

## 4. Discussion

This study aimed to identify demographic, personal, and workplace-related variables predicting job satisfaction among nurses. Concerning demographic and personal variables (age, gender, marital status, and personality traits), it was observed that health status and the personality traits of neuroticism, agreeableness, conscientiousness, and extraversion are correlated with job satisfaction among nurses. However, only age, health status, and neuroticism have been established as significant predictors of job satisfaction among nurses.

The participants’ age has been identified as a positive predictor of job satisfaction among nurses, indicating that older age contributes to higher job satisfaction. Results from other studies have indicated diverse findings, with some showing that age has no significant relationship with job satisfaction [41], while others suggested it could have a more substantial influence [42,43,44,45,46]. Possible reasons for such results may be the high expectations of younger nurses, which can be modified by the work environment that does not meet their expectations and diminishes with age [47]. Lowered expectations and increased work experience later in one’s career can result in greater job satisfaction, as individuals may find it easier to fulfill their expectations. Nurses, as they advance in their careers, may cultivate more positive sentiments toward their work. Furthermore, throughout their tenure, nurses have the opportunity to pursue additional education, opening doors to improved job prospects and often higher compensation. Holding a more favorable position and feeling adequately rewarded for one’s contributions can positively influence job satisfaction.

It was unsurprising that health status is positively associated with job satisfaction among nurses. In other words, the better the health status, the more satisfied nurses are with their jobs. These findings are consistent with other studies [4,48]. Understandably, nurses with poorer health are less satisfied with their jobs. The nursing profession can be physically challenging at times, with exhausting long shifts, night work, or an excessive patient load per nurse. The inability to perform job duties fully or at the cost of additional physical suffering will undoubtedly lead to a negative job experience. This, in turn, might contribute to a more negative perception of their job and workplace, ultimately influencing their level of job satisfaction. Also, healthier individuals generally experience an elevated quality of life and well-being, a decreased susceptibility to illness and injury, improved workplace productivity, and a greater propensity to make positive contributions to their workplace organization. This stands in contrast to individuals whose well-being is suboptimal [4,48].

Regarding the correlation between job satisfaction and personality traits, it has been shown that extraversion, agreeableness, and conscientiousness are positively associated, while neuroticism is negatively associated with job satisfaction. These results are consistent with the findings of other studies [23,24]; however, in those studies, it was shown that the personality trait of openness is also correlated with job satisfaction. It is important to mention that the studies did not involve a sample of nurses, which might be a potential factor contributing to the variation in results. It has been demonstrated that nurses who exhibit characteristics of extraversion, positive emotions, assertiveness, sociability, and conscientiousness, who approach their work with diligence, leading to a sense of accomplishment and satisfaction, as well as those who exbibit agreeableness, who are pleasant in interactions, resolve conflict through communication, express empathy, and help others, are more satisfied with their jobs.

However, concerning the relationship between extraversion, agreeableness, and conscientiousness with job satisfaction, it is lower than expected, and these traits did not emerge as significant predictors of job satisfaction among nurses.

On the other hand, neuroticism, as anticipated, is negatively linked to job satisfaction, and it exhibited a notable negative impact on job satisfaction among nurses. This result is in line with findings from various studies indicating that a prominent neuroticism trait is correlated and significantly contributes to lower job satisfaction [25,26]. However, it should be noted that these studies were not conducted with a sample of nurses.

Individuals with high scores on the neuroticism scale tend to be tense, anxious, and more susceptible to stress compared to those with lower scores [49], and they tend to perceive interpersonal relationships more negatively, even though they may not objectively be in line with this perception [50]. However, they also tend to create more negative interpersonal relationships due to their involved negative behaviors [51]. The result of our study might be a consequence of this. The nursing profession is a helping profession that involves interactions with other people, particularly in hospital environments, and requires teamwork. Consistent with the previously mentioned finding, nurses with a pronounced trait of neuroticism may tend to experience and create poorer interpersonal relationships with colleagues and patients alike.

Experiencing and creating negative interactions with patients and colleagues can result in frustration, heightened stress, and tension in professional relationships. This has been previously identified in research as a significant negative contributor to job satisfaction [49].

The mentioned results related to neuroticism also validate the core self-evaluation model. According to this model, four core self-evaluations influence an individual’s predisposition to job satisfaction, including general self-efficacy, locus of control, and neuroticism [33,34]. The findings indicate that higher neuroticism, indicating emotional instability, is linked to a reduced inclination for job satisfaction. This highlights the significance of comprehending an individual’s core self-evaluations, such as neuroticism, in predicting their overall satisfaction within the workplace. When discussing the locus of control, one possible explanation for this result, as shown in research, is that neuroticism is associated with an external locus of control [49]. Individuals with an external locus of control, believing that their actions do not significantly influence future outcomes, as they perceive results to be dependent on factors beyond their control [52], may resonate with this sense of powerlessness and the perception that the outcomes of their performance are determined by factors outside their control. This can lead to job dissatisfaction as they may feel powerless to improve their situation or achieve desired results, which can affect their overall perception of the work environment and work motivation.

Additionally, considering the correlations between neuroticism and organizational support, it has been demonstrated that there is a negative correlation between neuroticism and all three dimensions of organizational justice. The more pronounced the trait of neuroticism, the lower the level of distributive, interactional, and procedural justice. These findings are consistent with previous results, which have suggested a negative correlation between the mentioned constructs [53,54,55]. Possible reasons for these results are that more neurotic nurses tend to experience negative emotions and perceive unfair treatment compared to others [56], meaning they may perceive their superiors’ behavior towards them as unjust. It is important to consider the dynamic nature of the relationship between neuroticism and perceptions of organizational justice. It is possible that neurotic individuals, due to their tendency to experience negative emotions and perceive unfair treatment, may inadvertently contribute to a cycle of negative interactions and perceptions within the workplace. This could create a self-perpetuating cycle where their own neurotic tendencies lead to interpretations of injustice, which in turn further reinforce their negative emotions and perceptions.

The previously mentioned finding could suggest that neuroticism as a personality trait is undesirable in the nursing profession; individuals with more pronounced neuroticism were not inclined toward behaviors that involved helping others or were prosocial [52,57]. However, recent research has shown that in situations where helping others requires less social interaction or induces less anxiety, the negative association between neuroticism and prosocial behavior may disappear [58]. On the contrary, in these situations, the need to assist another person may evoke more compassion and care in individuals with pronounced neuroticism, leading them to engage in more prosocial behavior [59,60].

In future research, it would be interesting to explore whether patients on units where nurses exhibit a pronounced trait of neuroticism perceive the quality of healthcare as lower. Previous research has demonstrated the negative impact of this personality trait on job performance, and it is crucial to examine how this trait influences the satisfaction of recipients of nursing services, specifically whether neuroticism among nurses diminishes the perceived quality of nursing care.

It is important to note the results of the correlations between other personality traits and dimensions of organizational justice. Extraversion has been found to be associated with distributive and procedural justice, while it did not show a significant correlation with interactional justice. That is, the more pronounced the personality trait of extraversion, the higher the levels of distributive and procedural justice. The results are not in line with previous research that suggested no correlation between these constructs, but they were not conducted on a sample of nurses [54]. Possible reasons for these results are that extraverted nurses tend to more commonly support fair distribution and the protection of their own interests, which may result in more positive perceptions of distributive justice, as well as their tendency for active participation in processes and openness to procedures perceived as fair. An interesting finding is that there is no correlation between extraversion and interactional justice, although extraverted individuals tend to be sociable [56]. This does not necessarily mean that they will perceive all interpersonal relationships within the organization as fair or that they will be less sensitive to unfair treatment or lack of respect. Therefore, the relationship between extraversion and interactional justice may vary depending on individual characteristics and the context of the work environment.

Agreeableness is also associated with all three dimensions of organizational justice, meaning that the more pronounced the trait of agreeableness, the higher the level of all three dimensions of organizational justice. The results are consistent with previous research [55]. Individuals with a pronounced trait of agreeableness are inclined towards cooperation, altruism, and social sensitivity [56]; they usually trust others and feel guilty if they are advantaged compared to others [61,62]. It is possible that all of the above may result in more positive perceptions of organizational justice. That is, nurses who are more inclined towards cooperation may tend to advocate more for the fair distribution of resources (distributive justice), respect and courteous communication with others (interactional justice), and transparent and fair procedures in the organization (procedural justice). However, there is also a possibility that nurses who have a more pronounced trait of agreeableness will indeed be treated with respect and dignity by their superiors because agreeable individuals have a tendency to be selfless and flexible [56].

One of the primary contributions of this study is the identification of specific work-related factors as significant indicators of job satisfaction. The regression analysis indicated that job-related variables included in the third step contributed to explaining 12.8% of the variance in job satisfaction among nurses, while demographic variables contributed 6.9%, and personality traits contributed 8% of the explained variance. This suggests that the abovementioned variables have the greatest contribution to nurses’ job satisfaction in this study.

Regarding the correlation of job-related variables, all three dimensions of organizational justice have shown a significant correlation with job satisfaction. The correlation results between all three aspects of organizational justice and job satisfaction align with prior research [15,55]. They suggest that when nurses perceive a higher level of fairness in interpersonal treatment during the execution of organizational procedures when the decision-making process in their work organization is fair, and when they feel adequately rewarded for their contributions, their job satisfaction tends to be higher.

However, the correlation with interactional justice was lower than expected and did not prove to be a significant predictor of job satisfaction among nurses, and only two aspects, distributive and procedural justice, emerged as significant predictors. These findings align with prior research on perceived organizational justice, indicating that distributive justice and procedural justice were also significant predictors of job satisfaction [63,64,65,66].

While one might initially think that salary is the primary factor for nurses to be satisfied with their jobs, the situation is different. Distributive justice encompasses a much broader scope than just salary and refers to receiving fair rewards that take into account responsibilities, education level, effort, stress, job-related tension, and meeting these obligations [14].

Also, in cases where nurses perceive the decision-making process in their work organization as fair, their job satisfaction is higher. The reasons for such results likely can be found in the fact that nurses who perceive the decision-making process as fair are more inclined to accept organizational decisions, whether they feel negatively or positively toward them [67].

Considering the combined outcomes related to distributive and procedural justice, it is clear that a nurse’s perception of the organization’s decision-making process, along with feeling adequately rewarded for their contributions, encompassing not only monetary aspects but also various incentives, career growth opportunities, and social recognition, can positively influence job satisfaction. This understanding allows organizations to focus on improving these aspects, thereby positively influencing job satisfaction and potentially enhancing overall employee well-being and performance.

This study’s significant contribution lies in its illumination of the connections between organizational and personal mechanisms that impact job satisfaction among nurses. While past research concentrated on examining singular factors contributing to job satisfaction, this study sheds light on how job satisfaction is shaped by a myriad of diverse factors, intricately interacting to evoke this sensation.

However, it is important to note that this research cannot establish cause and consequence. Considering the previously presented findings, individuals with a strong disposition toward neuroticism may tend to perceive their surroundings more negatively, while those inclined towards positive affectivity tend to assess the same situations more optimistically.

Despite considerations of cause and effect, these results unequivocally emphasize that work-related variables and personal variables significantly predict job satisfaction in comparison to personal factors.

Initiatives within the organization that focus on improving distributive and procedural justice, while also considering factors such as health status and age, play a crucial role in fostering a positive workplace environment and contribute to enhancing job satisfaction among nurses. However, the presence of individual personality traits, such as neuroticism, introduces potential challenges that organizations must carefully address to achieve a more thorough comprehension of employee satisfaction.

## 5. Conclusions

This study revealed that, among personal variables, job satisfaction is associated with health status and personality traits such as neuroticism, extraversion, conscientiousness, and agreeableness, while among job-related variables, it is linked to all three dimensions of organizational justice: distributive, interactional, and procedural. The strongest predictors of job satisfaction among nurses were found to be health status, the personality trait of neuroticism, and distributive and procedural justice, with the age of nurses being slightly less powerful but still significant.

## 6. Study Limitations

There are several limitations that need to be considered when interpreting our findings. Firstly, the study is partly correlational in nature, so there is the possibility of reverse relationships between constructs. Additionally, the research was conducted in a single healthcare institution; therefore, it would be beneficial to investigate across multiple healthcare facilities and compare them to each other.

## Figures and Tables

**Table 1 behavsci-14-00235-t001:** Descriptive statistics of job satisfaction, organizational justice, and personality trait.

	M (Range)	SD
Job satisfaction	18.885 (7–25)	3.657
Distributive justice	12.503 (4–31)	4.439
Procedural justice	19.217 (7–35)	6.041
Interactional justice	33.670 (4–45)	8.547
Neuroticism	20.639 (6–41)	6.793
Extraversion	28.310 (12–39)	5.146
Openness	22.732 (11–40)	4.985
Agreeableness	29.397 (8–40)	5.261
Conscientiousness	35.695 (19–46)	4.889

Note: M—mean; SD—standard deviation.

**Table 2 behavsci-14-00235-t002:** Correlation between job satisfaction and demographic and personal variables.

		2.	3.	4.	5.	6.	7.	8.	9.
1. Job satisfaction	rho	0.102	−0.084	0.155 *	−0.219 **	0.156 *	−0.094	0.158 *	0.156 *
	*p* ^†^	0.199	0.288	0.049	0.005	0.049	0.233	0.045	0.048
2. Age	rho		−0.228 **	−0.483 **	0.003	−0.171 *	0.005	0.027	0.026
	*p* ^†^		0.004	<0.001	0.970	0.031	0.952	0.729	0.740
3. Gender	r			0.066	−0.215 **	0.044	0.175 *	−0.003	−0.107
	*p* ^‡^			0.406	0.006	0.577	0.026	0.969	0.177
4. Health status	rho				−0.054	0.090	−0.067	0.072	−0.029
	*p* ^†^				0.499	0.256	0.399	0.366	0.713
5. Neuroticism	rho					−0.387 **	−0.241 **	−0.354 **	−0.304 **
	*p* ^†^					<0.001	0.002	<0.001	<0.001
6. Extraversion	rho						0.221 **	0.273 **	0.347 **
	*p* ^†^						0.005	<0.001	<0.001
7. Openness	rho							0.078	0.009
	*p* ^†^							0.325	0.911
8. Agreeableness	rho								0.339 **
	*p* ^†^								<0.001
9. Conscientiousness	rho								-
	*p* ^†^								-

Note: ^†^ Spearman correlation; ^‡^ Point Biserial correlation; rho—Spearman correlation coefficient; r—Point Biserial correlation coefficient; *p*—statistical significance. * *p* < 0.05; ** *p* < 0.01.

**Table 3 behavsci-14-00235-t003:** Correlation between job satisfaction and job-related variables.

		2.	3.	4.	5.	6.
1. Job satisfaction	rho	−0.028	0.094	0.407 **	0.333 **	0.156 *
	*p* ^†^	0.720	0.236	<0.001	<0.001	0.049
2. Employment status	r		−0.483 **	0.067	0.167 *	0.187 *
	*p* ^‡^		<0.001	0.400	0.034	0.018
3. Job tenure	rho			−0.119	−0.303 **	−0.295 **
	*p* ^†^			0.134	<0.001	<0.001
4. Distributive justice	rho				0.422 **	0.274 **
	*p* ^†^				<0.001	<0.001
5. Procedural justice	rho					0.320 **
	*p* ^†^					<0.001
6. Interactional justice	rho					
	*p* ^†^					

Note: ^†^ Spearman correlation; ^‡^ Point Biserial correlation; rho—Spearman correlation coefficient; r—Point Biserial correlation coefficient; *p*—statistical significance. * *p* < 0.05; ** *p* < 0.05.

**Table 4 behavsci-14-00235-t004:** Correlation between personality traits and organizational justice.

		Distributive Justice	Procedural Justice	Interactional Justice
Neuroticism	rho	−0.189 *	−0.231 **	−0.225 **
	*p* ^†^	0.016	0.003	0.004
Extraversion	rho	0.226 **	0.251 **	0.095
	*p* ^†^	0.004	0.001	0.232
Openness	rho	−0.028	−0.001	−0.104
	*p* ^†^	0.728	0.992	0.190
Agreeableness	rho	0.238 **	0.166 *	0.178 *
	*p* ^†^	0.002	0.035	0.024
Conscientiousness	rho	0.105	0.095	0.151
	*p* ^†^	0.183	0.233	0.055

Note: ^†^ Spearman correlation; rho—Spearman correlation coefficient; *p*—statistical significance. * *p* < 0.05; ** *p* < 0.01.

**Table 5 behavsci-14-00235-t005:** Summary of the regression analysis.

					CI		
		β	t	*p*	Lower	Upper	AR^2^
1	(Constant)		5.143	<0.001	7.522	16.906	0.069 **
	Age	0.277	2.721	0.007 **	0.022	0.139	
	Gender	−0.047	−0.576	0.565	−2.105	1.154	
	Marital status—single	0.043	0.446	0.656	−1.193	1.889	
	Marital status—divorced	−0.058	−0.725	0.470	−2.653	1.229	
	Marital status—widower	−0.103	−1.312	0.192	−11.960	2.414	
	Health status	0.279	3.171	0.002 **	0.450	1.935	
2	(Constant)		3.624	<0.001	7.514	25.529	0.149 **
	Age	0.261	2.622	0.010 *	0.019	0.133	
	Gender	−0.076	−0.935	0.351	−2.397	0.857	
	Marital status—single	0.037	0.389	0.698	−1.200	1.787	
	Marital status—divorced	−0.097	−1,232	0.220	−3.067	0.711	
	Marital status—widower	−0.091	−1,167	0.245	−11.415	2.939	
	Health status	0.235	2.741	0.007 **	0.280	1.724	
	Neuroticism	−0.267	−2.978	0.003 **	−0.239	−0.048	
	Extraversion	0.132	1.507	0.134	−0.029	0.217	
	Openness	−0.156	−1.945	0.054	−0.231	0.002	
	Agreeableness	−0.002	−0.028	0.978	−0.116	0.112	
	Conscientiousness	−0.004	−0.051	0.959	−0.127	0.121	
3	(Constant)		2,441	0.016	2.246	21.361	0.277 **
	Age	0.192	0.611	0.542	−0.125	0.237	
	Gender	−0.065	−0.855	0.394	−2.162	0.857	
	Marital status—single	0.019	0.217	0.829	−1.265	1.576	
	Marital status—divorced	−0.105	−1.439	0.152	−3.043	0.479	
	Marital status—widower	−0.062	−0.848	0.398	−9.631	3.848	
	Health status	0.230	2.898	0.004 **	0.312	1.653	
	Neuroticism	−0.195	−2.312	0.022 *	−0.195	−0.015	
	Extraversion	0.071	0.864	0.389	−0.065	0.166	
	Openness	−0.106	−1.394	0.165	−0.188	0.033	
	Agreeableness	−0.054	−0.681	0.497	−0.146	0.071	
	Conscientiousness	−0.014	−0.184	0.854	−0.126	0.104	
	Distributive justice	0.273	3.535	0.001 **	0.099	0.351	
	Procedural justice	0.180	2.221	0.028 *	0.012	0.206	
	Interactional justice	0.064	0.824	0.411	−0.038	0.093	
	Employment status	−0.009	−0.111	0.912	−1.456	1.301	
	Job tenure	0.130	0.438	0.662	−0.129	0.203	

Note: *p*—statistical significance; β—regression coefficient; t—the size of the difference relative to the variation in your sample data; AR^2^—coefficient of determination; CI—confidence interval. * *p* < 0.05; ** *p* < 0.01.

## Data Availability

The data are available from the corresponding author upon reasonable request.

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
