# Peer review of "Influence of Personality Traits and Organizational Justice on Job Satisfaction among Nurses"

_behavsci, 2024, doi:10.3390/bs14030235_

Round 1

Reviewer 1 Report

Comments and Suggestions for Authors

Dear Author(s),

Thank you for the submission.

This topic is relevant. It is better to include a section "Literature Review" to support your research.

All the best.

Reviewer 2 Report

Comments and Suggestions for Authors

Thank you for inviting me to review. I congratulate the authors who decided to raise an important issue in the work of nurses.

Interesting work, written in a logical way. The presented results are adequate to the title and purpose of the work, although one could consider the title of the manuscript - whether influence or connection can be studied „The contribution of personality traits and organizational justice to job satisfaction among nurses”.

The size of the sample is unquestionable. The list of references is up-to-date and correct, advanced statistical methods are used, thorough analysis of the literature, in-depth discussion. Very neat and clear graphic design. The work is suitable for publication after very minor corrections.

The authors did not avoid minor shortcomings. I hope that your comments will improve the manuscript:

1. research procedure: - requires supplementation in terms of the organization and course of the study - the method of distributing research tools among the respondents, the method of ensuring anonymization of data, the criteria for including and excluding participants from the study and obtaining consent to use the research tools in the presented study.

2. limitations should be included in the discussion.

3. the article should end with clearly formulated conclusions, which should also be included in the summary.

4. it is important not to overinterpret the relationship as a causal relationship, because not every result resulting from the analysis of the relationship between two variables is related to them, and must be of a cause-and-effect nature, e.g.

Lines 299-303 „Regression analysis indicated that these variables have the greatest impact on job satisfaction among nurses in this study. All three dimensions of organizational justice have shown a significant correlation with job satisfaction. The correlation results between all three aspects of organizational justice and job satisfaction align with prior research [57, 15]”.

Reviewer 3 Report

Comments and Suggestions for Authors

This is an interesting cross-sectional study of personality and organizational traits and their impact on nurses' job satisfaction. The background and discussion do a nice job giving an overview of the existing literature and what might be unique or of especial relevance to nurses, and what changes in the workplace might improve satisfaction. The findings are not surprising and mostly align with what has been found previously in the literature more generally, but the focus on nurses but empiric evidence from recent survey is a welcome addition to the literature.

The one big thing that felt like it was missing, and that would enhance the value of the paper if it were statistically viable to include, would be analyses looking at the interaction of personality and organizational factors. For example, it would be very interesting to see if there is any correlation between personality factor ratings and ratings of organizational justice. The implication of this is important. As noted in the paper, individuals with greater neuroticism and/or external locus of control tend to not feel they have control. I can imagine this might translate into a general skepticism towards or lowered perception of organizational justice, regardless of objective conditions. In any case, the interaction of personality and organizational factors is of great interest, but is not discussed. I would encourage the authors to revise to whatever extent it is possible to include discussion of personality-organizational factor interactions and consider these kinds of issues. 

The writing is clear and there are just a couple of sentences that were a bit confusing:

- line 233: "employees who prioritize their health and well-being" - this makes it sound like people get to choose whether they are healthy or not. Recognizing there is interplay between self-perception of health/well-being and externally measured health/well-being, nevertheless this sentence sounds both a bit insensitive to employees with health challenges, and confusing. I suggest rewording to clarify better what is meant.

- lines 279-281 discuss the issue of external locus of control and state that "more neurotic nurses perceive people who think they have no control over external events" - this may be my lack of understanding of external locus of control, but I think you should clarify for the reader if those with an external locus of control believe that this holds for all people (ie, no one has control). I have thought that those with external locus of control often assign control to others in their minds - if that is so, this sentence is confusing.

 Finally a note for copy editors: whatever the journal style is, regarding commas or periods to mark decimal places, there is a need to use it consistently. Some p values are reported with a comma, some with a period, and all other values are reported with a comma.

Comments on the Quality of English Language

The English is fine - sometimes simpler than one might expect in academic writing, but that is actually a good thing. Aside from a few confusing sentences where language might be a contributing factor, the writing is clear and correct.
